# Action Recognition via Adaptive Semi-Supervised Feature Analysis

Zengmin Xu [1,2,3,†] , Xiangli Li [1,2,†] , Jiaofen Li [1,2,*] , Huafeng Chen [4,*] and Ruimin Hu [5]

1 School of Mathematics and Computing Science, Guangxi Colleges and Universities Key Laboratory of Data Analysis and Computation, Guilin University of Electronic Technology, Guilin 541004, China; xzm@guet.edu.cn (Z.X.)
2 Center for Applied Mathematics, Guangxi (GUET), Guilin 541004, China
3 Anview.ai, Guilin 541010, China
4 School of Computer Engineering, Jingchu University of Technology, Jingmen 448000, China
5 National Engineering Research Center for Multimedia Software, School of Computer Science, Wuhan University, Wuhan 430072, China
* Correspondence: lixiaogui1290@163.com (J.L.); chenhuafeng@jcut.edu.cn (H.C.)
† These authors contributed equally to this work.

**Abstract:** This study presents a new semi-supervised action recognition method via adaptive feature analysis. We assume that action videos can be regarded as data points in embedding manifold subspace, and their matching problem can be quantified through a specific Grassmannian kernel function while integrating feature correlation exploration and data similarity measurement into a joint framework. By maximizing the intra-class compactness based on labeled data, our algorithm can learn multiple features and leverage unlabeled data to enhance recognition. We introduce the Grassmannian kernels and the Projected Barzilai–Borwein (PBB) method to train a subspace projection matrix as a classifier. Experiment results show our method has outperformed the compared approaches when a few labeled training samples are available.

**Keywords:** non-monotone line-search; two-point step-size gradient; Grassmannian kernels

## 1. Introduction

Effective feature representation is key to image processing [1–3] and video understanding [4–6]. Spatio-temporal features [7,8], subspace features [9,10], and label information [11,12] have been investigated for action recognition. Nevertheless, in Figure 1, we observe that video understanding represents a significant evolution through new datasets and approaches. The activities scenarios have moved on from simple sports, isolated movies, normal surveillance to cluttered home sequences, egocentric interactions of kitchens, real-world anomalous events, part-level action parsing, dark environments, and complex surveillance videos. Considering the various views, illumination, poses, and outdoor conditions of activities, while the data distribution of feature space remains uncertain, how do we discover the underlying embedded subspace for different types of features, and what are the boundaries of action clips?

On the other hand, large-scale videos are constantly emerging nowadays; thus, lots of segments need automatic labeling, but this requires human labor. Large amounts of normal behaviors are more than those of anomalous events. It is important to measure data similarity by sample matching with distance metric learning. Noticeably, some segments in untrimmed videos may be out of specific categories [13] , or there are no annotations of new sequences in the dark environment [14]. Therefore, in order to solve the point-matching problem in a semi-supervised manner, we discuss how to convert the video-set matching problem to a data distance measurement problem on the manifold subspace.

Correlations between multiple features may provide distinctive information; hence, feature correlation mining has been explored to improve the recognition results when

labeled data are scarce [10,15]. However, these approaches may have limitations in learning discriminant features. First, although existing algorithms evaluate the common shared structures among different actions, they do not take inter-class separability into account. Second, current semi-supervised approaches solve the non-convex optimization problem by impressive derivation, but the global optimum may not be computed mathematically through the alternating least-squares (ALS) iterative method.

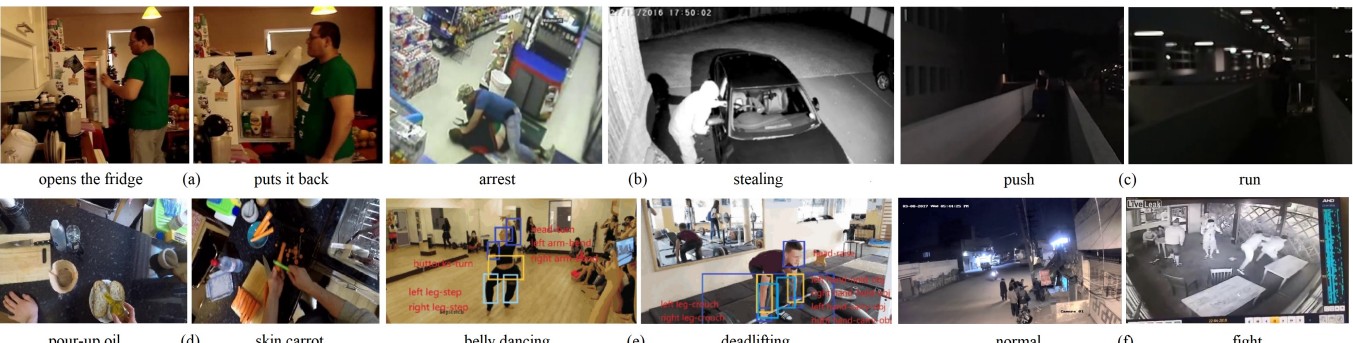

**Figure 1.** Sample frames from (**a**) home activities (Charades [16]), (**b**) real-world anomalies (UCF-Crime [13]), (**c**) dark environments (ARID [14]), (**d**) egocentric interactions (EPIC-KITCHENS-100 [17]), (**e**) part-level actions (Kinetics-TPS [18]), and (**f**) fight scenarios (CSCD [19]). All videos have a large gap towards target-oriented and diversity-oriented. (**a**) Charades depicts cluttered home actions from multimedia, (**b**) UCF-Crime shows real-world events containing anomalous and normal segments in untrimmed videos, (**c**) ARID aims to recognize actions in low illumination through semi-supervised methods, (**d**) EPIC-KITCHENS-100 consists of daily activities in the kitchen from first-person videos, (**e**) Kinetics-TPS develops a large-scale kinetics-temporal part state for encoding the composition of body parts, and (**f**) CSCD collects fight and no-fight scenarios from surveillance cameras.

## 2. Motivation and Contributions

To overcome the limitations of using multiple features for training, we propose modeling intra-class compactness and inter-class separability simultaneously, then capturing high-level semantic patterns via multiple-feature analysis. Considering the optimization process, we introduce the PBB algorithm because of its effectiveness in obtaining an optimal solution [20]. The PBB method is a non-monotone line-search technique considered for the minimization of differentiable functions on closed convex sets [21].

Inspired by the research using multiple features [11,15], our framework was extended in a multiple-feature-based manner to improve recognition. We proposed the characterization of high-level semantic patterns through low-level action features using multiple-feature analysis. Multiple features were extracted from different views of labeled and unlabeled action videos. Based on the constructed graph model, pseudo-information of unlabeled videos can be generated by label propagation and feature correlations. For each type of feature, nearby samples preserve the consistency separately, while unlabeled training data perform the label prediction by jointly global consistency of multiple features. Thus, an adaptive semi-supervised action classifier was trained. The main contributions can be summarized as follows:

(1) This work first simultaneously considers manifold learning and Grassmannian kernels in semi-supervised action recognition, as we assume that action video samples may be found in a Grassmannian manifold space. By modeling an embedding manifold subspace, both inter-class separability and intra-class compactness were considered.

(2) To solve the unconstrained minimization problem, we incorporate the PBB method to avoid matrix inversion, and apply globalization strategy via adaptive step sizes to render the objective functions non-monotonic, leading to improved convergence and accuracy.

(3) Extensive experiments verified that our method is better than other approaches on three benchmarks in a semi-supervised setting. We believe that this study presents valuable insights into adaptive feature analysis for semi-supervised action recognition.

## 3. Related Work

We review the related research on semi-supervised action recognition, multiple-feature analysis, and embedded subspace representation in this section.

### 3.1. Semi-Supervised Action Recognition

Unlabeled samples are valuable for learning data correlations in a semi-supervised manner [9,10,12,22]. Although they tend to achieve remarkable performance via semi-supervised learning with limited labeled data, there are still many issues, such as inherent multi-modal attributes leading to local optimum, or unconvincing pseudo-labels leading to inaccurate predictions [23,24].

Si et al. [25] tackle the challenge of semi-supervised 3D action recognition for effectively learning motion representations from unlabeled data. Singh et al. [6] maximize the similarity of the same video at two different speeds, and recognize actions by training a two-pathway temporal contrastive model. Kumar and Rawat [26] develop a spatio-temporal consistency-based approach with two regularization constraints: temporal coherency and gradient smoothness, which can detect video action in an end-to-end semi-supervised manner.

### 3.2. Multiple-Feature Analysis

Because we can describe an object by different features that provide different discriminative information, multiple-feature analysis has gained increasing interest in many applications. In the early and late fusion strategies, multi-stage fusion schemes have recently been investigated [10,27–29]. However, the correlations of each feature type have not been considered in most late-fusion approaches.

Wang et al. [10] apply shared structural analysis to characterize discriminative information and preserve data distribution information from each type of feature. Chang and Yang [15] discover shared knowledge from related multi-tasks, take various correlations into account, then select features in a batch mode. Huynh-The et al. [30] capture multiple high-level features at image-based representation by fine-tuning a pre-trained network, transfer the skeleton pose to encoded information, and depict an action through spatial joint correlations and temporal pose dynamics.

### 3.3. Embedded Subspace Representation

Previous studies have shown that manifold subspace learning can mine geometric structure information by considering the space of probabilities as a manifold [31–33]. Recent research focuses on graph-embedded subspace or distance metric learning to measure activity similarity [34–38].

Rahimi et al. [39] build neighborhood graphs with geodesic distance instead of Euclidean distance, and project high-dimensional action to low-dimensional space by kernelized Grassmann manifold learning. Yu et al. [40] propose an action-matching network to recognize open-set actions, construct an action dictionary, and classify an action via the distance metric. Peng et al. [41] alleviate the over-smoothing issue of graph representation when multiple GCN layers are stacked by the flexible graph deconvolution technique.

The two aforementioned studies [9,10] are similar to ours. They assume that the visual words in different actions share a common structure in a specific subspace. A transformation matrix is introduced to characterize the shared structures. They solve the constrained non-convex optimization problem through an ALS–like iterative approach and matrix derivation. Nevertheless, the deduced inverse matrix is poorly scaled during optimization or close to singular, which may lead to inaccurate results.

To address these problems, we hypothesize that manifold mapping can preserve the local geometry and maximize discriminatory power, as shown in Figure 2. However, we did not aim to mine shared structures. Therefore, we ignored shared-structure regularization and modeled the manifold by creating two graphs. As the optimization solution in [9,10] may be mathematically imprecise, Karush–Kuhn–Tucker (KKT) conditions and PBB are introduced to improve algorithm convergence and avoid matrix inversion.

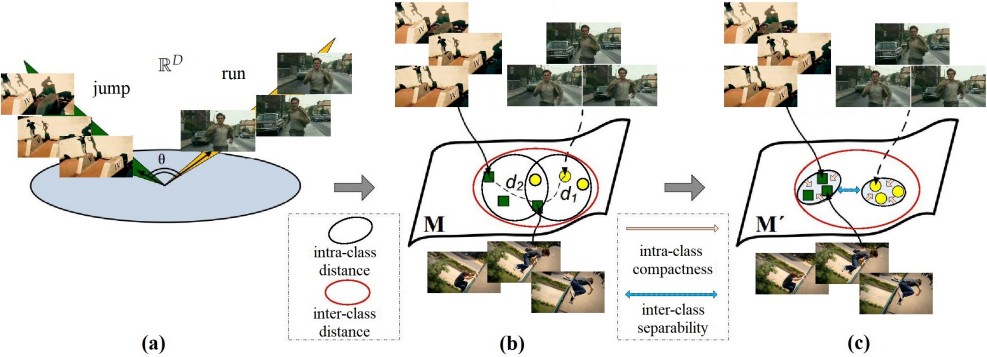

**Figure 2.** An illustration of our method. (**a**) Video sets can be represented in $\mathbb{R}^D$. We can use the principal angles between them to compare two actions. (**b**) Data points on the Grassmannian manifold $M$ can be described as linear subspaces in $\mathbb{R}^D$. When points on the manifold have a proper geodesic distance, the video-set-matching problem may be converted to a points distance measurement problem. (**c**) By employing a proper Grassmannian kernel, data points can be mapped into another Grassmannian manifold $M'$ where the same actions become closer while different actions are well separated.

Different from another related research [12], this work makes two major modifications as follows: multiple-feature analysis with combined Grassmannian kernels, and non-monotone line-search strategy with adaptive step sizes.

## 4. Proposed Approach

Our approach incorporates several techniques, including semi-supervised action recognition, multiple-feature analysis, PBB, KKT, manifold learning, and Grassmannian kernels. It is named Kernel Grassmann manifold analysis (KGMA).

### 4.1. Formulation

To leverage the multiple-feature correlation, $n$ training sample points $\mathbb{X} = [X_1, ..., X_n] \in \mathbb{R}^{d \times n}$ are defined from the underlying Grassmannian manifold, where $X_i \in \mathbb{R}^{d \times 1}$. We aim to uncover a new manifold while preserving the local geometry of data points, that is, $\alpha : X_i \to F_i$. Since we should demonstrate data distribution on the manifold, a predicted label matrix $\mathbb{F} = [F_1, ..., F_n] \in \mathbb{R}^{n \times n}$ is defined, where the predicted vector of the $i$-th datum $X_i \in \mathbb{X}$ is $F_i \in \mathbb{R}^{n \times 1}$.

We assume that a similarity measurement of data points in the manifold subspace is available through a Grassmannian kernel [31] $k_{i,j} = \langle X_i, X_j \rangle$. By confining the solution to a linear function, that is, $\alpha_i = \sum_{j=1}^{n} a_{ij} X_j$, we define the prediction function $f$ as $f(X_i) = F_i = (\langle \alpha_1, X_i \rangle, \langle \alpha_2, X_i \rangle, ..., \langle \alpha_r, X_i \rangle)^T$. By denoting $A_l = (a_{l1}, ..., a_{ln})^T$ and $K_i = (k_{i1}, ..., k_{in})^T$, it can be shown that $\langle \alpha_l, X_i \rangle = A_l^T K_i$, and thus, $f(\mathbb{X}) = \mathbb{F} = \mathbb{A}^T \mathbb{K} \approx \mathbb{Y}$, where $\mathbb{A} = [A_1|A_2|...|A_r]$ and $\mathbb{K} = [K_1|K_2|...|K_n]$. As mentioned in [42], the performance of the least-square loss function is comparable to hinge loss or logistic loss. This is associated with its diagonal matrix $\mathbb{Y} = [Y_1, ..., Y_n] \in \{0, 1\}^{n \times n}$, where $Y_i \in \{0, 1\}^{n \times 1}$ is the label matrix. We employ least-squares regression to solve the following optimization problem, then obtain the projection matrix $\mathbb{A}$:

$$\min_{\mathbb{A}} \|\mathbb{A}^T \mathbb{K} - \mathbb{Y}\|_F^2 + \eta \|\mathbb{A}^T\|_F^2, \tag{1}$$

where $\eta$ is the regularization parameter. $\| \cdot \|_F^2$ denotes the Frobenius norm. $\|\mathbb{A}^T\|_F^2$ controls the model complexity to prevent overfitting.

### 4.2. Manifold Learning

In contrast to [10], which utilizes a graph model to estimate data distribution on the manifold, we model the local geometrical structure by generating between-class similarity graph $G_b$ and within-class similarity graph $G_w$, where $G_w(i,j) = 1$, if $x_i \in N_w(x_j)$ or $x_j \in N_w(x_i)$; otherwise, $G_w(i,j) = 0$. $G_b(i,j)$ applies the same method, although it selects $x_i \in N_b(x_j)$ or $x_j \in N_b(x_i)$, where $N_b(x_i)$ contains neighbors with different labels, and $N_w(x_j)$ is the set of neighbors $x_j$ sharing the same label as $x_i$. Notably, the intra-class and inter-class distances can be mapped on a manifold by similarity graphs [33].

Inspired by manifold learning [12,31,33], we maximize inter-class separability and minimize intra-class compactness simultaneously. An ideal transform pushes the connected points of $A_b$ to the extent possible while moving the connected points of $A_w$ closer. The discriminative information can be represented as follows:

$$
\begin{aligned}
f &= \frac{1}{2}\sum_{i,j=1}^{n}(F_i - F_j)^2 G_w(i,j) - \frac{1}{2}\beta\sum_{i,j=1}^{n}(F_i - F_j)^2 G_b(i,j) \\
&= tr(\mathbb{F}^T(L_w - \beta L_b)\mathbb{F}),
\end{aligned}
\tag{2}
$$

where $\beta$ is a regularization parameter, which controls the trade-off between inter-class separability and intra-class compactness. $tr(\cdot)$ denotes the trace operator, and $L_w = D_w - G_w$ denotes the Laplacian matrix. Furthermore, $D_b$ is a diagonal matrix with $D_b(i,i) = \sum_{j=1}^{n} G_b(i,j)$, and $D_w$ is a diagonal matrix with $D_w(i,i) = \sum_{j=1}^{n} G_w(i,j)$.

### 4.3. Multiple-Feature Analysis

Multiple features imply combining kernelized embedding features, data-point manifold subspace learning (1st term in Equation (4)), and label propagation (2nd term in Equation (4)) with low-level feature correlations (3rd term in Equation (4)) for labeled and unlabeled data.

We modify the aforementioned function to leverage both labeled and unlabeled samples. First, the training dataset is redefined as $\mathbb{X} = [\mathbb{X}_l^T, \mathbb{X}_u^T]^T$, where $\mathbb{X}_l = [X_1, ..., X_m]^T$ is the labeled data subset, and $\mathbb{X}_u = [X_{m+1}, ..., X_n]^T$ is the unlabeled data subset. The label matrix $\mathbb{Y} = [\mathbb{Y}_l^T, \mathbb{Y}_u^T]^T$, where $\mathbb{Y}_l = [Y_1, ..., Y_m]^T \in \{1\}^{m \times m}$. The unlabeled matrix $\mathbb{Y}_u = [Y_{m+1}, ..., Y_n]^T \in \{0\}^{(n-m) \times (n-m)}$. According to [9,43], diagonal label matrix $\mathbb{Y}$ and the similarity graphs $G_w, G_b$ should be consistent with the label prediction matrix $\mathbb{F}$. We generalize the graph-embedded label consistency as follows:

$$
\min_{\mathbb{F}} tr(\mathbb{F}^T(L_w - \beta L_b)\mathbb{F}) + \|\mathbb{F} - \mathbb{Y}\|_F^2,
\tag{3}
$$

In contrast to previous shared-structure learning algorithms, we do not consider shared-structure learning within a semi-supervised learning framework. Alternatively, we propose a novel joint framework that incorporates the multiple-feature analyses of multiple manifolds. As discussed in the problem formulation section, by employing the Frobenius norm regularized loss function, we can reformulate the objective:

$$
\begin{aligned}
\min_{\mathbb{F},\mathbb{A}} \ & tr(\mathbb{F}^T(L_w - \beta L_b)\mathbb{F}) + \|\mathbb{F} - \mathbb{Y}\|_F^2 \\
& + \mu\left(\|\mathbb{A}^T\mathbb{K} - \mathbb{Y}\|_F^2 + \eta\|\mathbb{A}^T\|_F^2\right),
\end{aligned}
\tag{4}
$$

where $\beta > 0$, $\mu > 0$ and $\eta > 0$ are regular parameters.

Equation (4) is an unconstrained convex optimization problem; hence, we can obtain the global optimum by performing ALS or the projected gradient method. Although the correlation matrix can only be singular under specific circumstances, the projected gradient

method can handle the aforementioned issues without matrix inversion [20], and therefore leads to a better optimum than ALS. Notably, the convergence conditions in [9,10] merely depend on a monotone decrease, which may result in mathematically improper convergence; therefore, KKT conditions are utilized to consider this problem.

*4.4. Grassmannian Kernels*

The similarity between two action sample points $X_i$ and $X_j \in \mathbb{R}^{d \times 1}$ can be measured by projective kernel combination:

$$k_{i,j}^{[proj]} = \| X_i^T X_j \|_F^2 . \tag{5}$$

One attempt to solve the point-matching problem is the notion of principal angles [31]. Given $X_i$ and $X_j$, we can define the canonical correlation kernel as

$$k_{i,j}^{[cc]} = \max_{a_p \in span(X_i)} \max_{b_q \in span(X_j)} a_p^T b_q, \tag{6}$$

subject to $a_p^T a_p = b_p^T b_p = 1$ and $a_p^T a_q = b_p^T b_q = 0, p \neq q$.

We create a combined Grassmannian kernel through existing Grassmannian kernels [31].

$$k^{[A+B]} = \delta^{[A]} k^{[A]} + \delta^{[B]} k^{[B]}, \tag{7}$$

where $\delta^{[A]}, \delta^{[B]} \geqslant 0$. Notably, $k^{[A]} + k^{[B]}$ defines a new kernel based on the theory of reproducing the kernel Hilbert space, as described in [31].

*4.5. Optimization*

According to [20,21], a general unconstrained minimization problem can be solved by the trace operator and the PBB method. Hence, a new objective function $g(\mathbb{F}, \mathbb{A})$ instead of Equation (4) is defined:

$$\begin{aligned} g(\mathbb{F}, \mathbb{A}) = & tr(\mathbb{F}^T (L_w - \beta L_b) \mathbb{F}) + tr(\mathbb{F} - \mathbb{Y})^T (\mathbb{F} - \mathbb{Y}) \\ & + \mu tr(\mathbb{A}^T \mathbb{K} - \mathbb{Y})^T (\mathbb{A}^T \mathbb{K} - \mathbb{Y}) + \mu \eta tr(\mathbb{A}\mathbb{A}^T). \end{aligned} \tag{8}$$

If $(\mathbb{F}^*, \mathbb{A}^*)$ is an approximate stationary point in Equation (8), it must satisfy the KKT conditions in Equation (8). Then, we have an iteration-stopping criterion

$$\|\nabla g_{\mathbb{F}}(\mathbb{F}^*, \mathbb{A}^*)\|^2 + \|\nabla g_{\mathbb{A}}(\mathbb{F}^*, \mathbb{A}^*)\|^2 \leqslant \varepsilon, \tag{9}$$

where $\varepsilon$ is a non-negative small constant.

*4.6. Projected Barzilai-Borwein*

Similar to [20], a sequence of feasible points $(\mathbb{F}^t, \mathbb{A}^t)$ is generated by the gradient method:

$$\begin{aligned} d\mathbb{F}^t = -\lambda^t \nabla g_{\mathbb{F}}(\mathbb{F}^t, \mathbb{A}^t), \quad \mathbb{F}^{t+1} = \mathbb{F}^t + \sigma_t d\mathbb{F}^t, \\ d\mathbb{A}^t = -\lambda^t \nabla g_{\mathbb{A}}(\mathbb{F}^t, \mathbb{A}^t), \quad \mathbb{A}^{t+1} = \mathbb{A}^t + \sigma_t d\mathbb{A}^t, \end{aligned} \tag{10}$$

where $\lambda^t = \min\{\lambda_{max}, \max\{\lambda_{min}, \lambda_{ABB}^t\}\} > 0$ is another step size, and $\sigma_t$ denotes the non-monotone line-search step size that is determined through an appropriate selection rule. Following [21], we have two choices for step size:

$$\begin{aligned} \lambda_{BB1}^{t+1} &= \frac{\langle s_1^t, s_1^t \rangle + \langle s_2^t, s_2^t \rangle}{\langle s_1^t, y_1^t \rangle + \langle s_2^t, y_2^t \rangle}, \\ \lambda_{BB2}^{t+1} &= \frac{\langle s_1^t, y_1^t \rangle + \langle s_2^t, y_2^t \rangle}{\langle y_1^t, y_1^t \rangle + \langle y_2^t, y_2^t \rangle}, \end{aligned} \tag{11}$$

where

$$s_1^t = \mathbb{F}^{t+1} - \mathbb{F}^t, \quad s_2^t = \mathbb{A}^{t+1} - \mathbb{A}^t,$$
$$y_1^t = \nabla g_{\mathbb{F}}(\mathbb{F}^{t+1}, \mathbb{A}^{t+1}) - \nabla g_{\mathbb{F}}(\mathbb{F}^t, \mathbb{A}^t), \tag{12}$$
$$y_2^t = \nabla g_{\mathbb{A}}(\mathbb{F}^{t+1}, \mathbb{A}^{t+1}) - \nabla g_{\mathbb{A}}(\mathbb{F}^t, \mathbb{A}^t),$$

The characteristic of the adaptive step sizes (11) can render the objective functions non-monotonic; hence, $g(\mathbb{F}^t, \mathbb{A}^t)$ may increase in some iterations. Alternatively, using (11) is better than merely using one of them [21]; the step size is expressed by

$$\lambda_{ABB}^t = \begin{cases} \lambda_{BB1}^t, & \text{for odd number } t \\ \lambda_{BB2}^t, & \text{for even number } t \end{cases} \tag{13}$$

To guarantee the convergence of $(\mathbb{F}^t, \mathbb{A}^t)$, a globalization strategy based on the non-monotone line-search technique is described as [20]

$$g(\mathbb{F}^{t+1}, \mathbb{A}^{t+1}) \leqslant C_t + \gamma \sigma_t \{ \langle \nabla g_{\mathbb{F}}(\mathbb{F}^t, \mathbb{A}^t), d\mathbb{F}^t \rangle \\ + \langle \nabla g_{\mathbb{A}}(\mathbb{F}^t, \mathbb{A}^t), d\mathbb{A}^t \rangle \} \tag{14}$$

where $\tau \in (0, 1]$, $C_t$ are the parameters of the Armoji line-search method [21]. Following [20], in order to overcome some drawbacks of non-monotone techniques, the traditional largest function value is converted by the weighted-average function value:

$$C_t = \frac{\tau \cdot \min\{t-1, M\} C_{t-1} + g(\mathbb{F}^t, \mathbb{A}^t)}{\tau \cdot \min\{t-1, M\} + 1}, \tag{15}$$

## 5. Experiments

The proposed method called KGMA is summarized in Algorithm 1. The conventional method that uses SPG [12] and the ALS method instead of PBB, called kernel spectral projected gradient analysis (KSPG) and kernel alternating least-squares analysis (KALS), respectively, was also adopted to solve the objective function (8) for comparison in our experiments.

---

**Algorithm 1:** Kernel Grassmann Manifold Analysis (KGMA).

**Input** : Training sample $\mathbb{X} \in \mathbb{R}^{d \times n}$
        Diagonal labels $\mathbb{Y} \in \{0, 1\}^{n \times n}$
        Semi-supervised parameters $\beta, \mu$ and $\eta$.
        The PBB parameters $M, \lambda_{\min}, \lambda_{\max}, \sigma_t, \gamma, \tau, C_t$
**Output:** Optimised $\mathbb{A}^* \in \mathbb{R}^{n \times n}$

    Grassmann matrix $[\mathbb{K}]_{ij}$ for all $X_i, X_j$
    Between-class similarity graph $L_b \in \mathbb{R}^{n \times n}$
    Within-class similarity graph $L_w \in \mathbb{R}^{n \times n}$
    Initialise $\mathbb{F}^0 \in \mathbb{R}^{n \times n}, \mathbb{A}^0 \in \mathbb{R}^{n \times n}$ randomly
    Initialise $C_0 = g(\mathbb{F}^0, \mathbb{A}^0)$
    Initialise $t = 0, \lambda^0 = 1, \sigma_0 = 1, \gamma = 0.1, \tau = 0.3$
    **repeat**                                                                      ▷ PBB Method

        **if** *(14) is satisfied* **then**
            Compute $\mathbb{F}^{t+1}, \mathbb{A}^{t+1}$ according to (10)
            Compute $s_1^t, s_2^t, y_1^t, y_2^t$ according to (12)
            **if** $\langle s_1^t, y_1^t \rangle + \langle s_2^t, y_2^t \rangle \leqslant 0$ **then** $\lambda^{t+1} = \lambda_{max}$;
            **else** $\lambda^{t+1} = \min\{\lambda_{max}, \max\{\lambda_{min}, \lambda_{ABB}^{t+1}\}\}$;
            $t = t + 1$
        **until** *Convergence according to (9)*;
    Return $\mathbb{A}^*$

---

### 5.1. Features

For handcrafted features, we follow [12] to extracted improved dense trajectories (IDTs) and Fisher vector (FV), as shown in Figure 3.

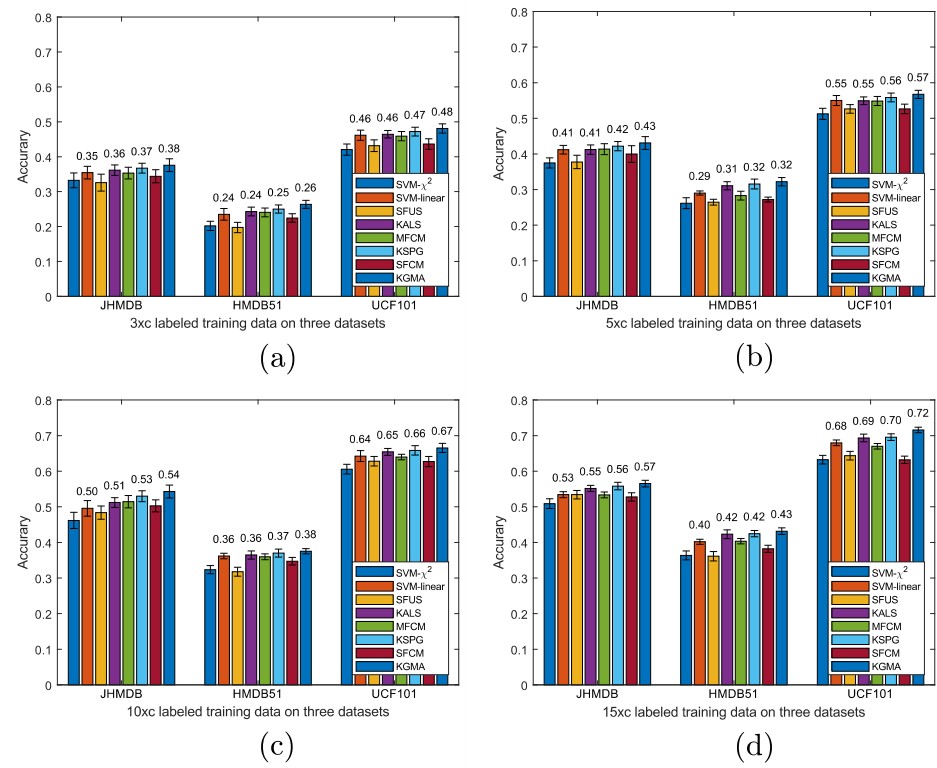

**Figure 3.** Comparison (average accuracy $\pm$ std) with IDT+FV when different number of training samples are labeled, gmmSize = 16.

For deep-learned features, we retrained the temporal segment network (TSN) [7] models of $15 \times c$, and then extracted the global pool features of $15 \times c$ using a pre-trained TSN model, concatenating rgb + flow into 2048 dimensions with power L2-normalization, as listed in Table 1.

**Table 1.** Comparison with deep-learned features (average accuracy $\pm$ std) when $15 \times c$ training videos are labeled

|              | **JHMDB**             | **HMDB51**            | **UCF101**            |
| ------------ | --------------------- | --------------------- | --------------------- |
| SFUS         | $0.6942 \pm 0.0121$   | $0.5217 \pm 0.0114$   | $0.7910 \pm 0.0087$   |
| SFCM         | $0.7125 \pm 0.0099$   | $0.5394 \pm 0.0108$   | $0.8070 \pm 0.0101$   |
| MFCU         | $0.7154 \pm 0.0088$   | $0.5556 \pm 0.0098$   | $0.8429 \pm 0.0085$   |
| SVM-$\chi^2$ | $0.6931 \pm 0.0106$   | $0.5190 \pm 0.0095$   | $0.8138 \pm 0.0108$   |
| SVM-linear   | $0.7140 \pm 0.0086$   | $0.5385 \pm 0.0077$   | $0.8450 \pm 0.0087$   |
| KSPG         | $0.7287 \pm 0.0114$   | $0.5697 \pm 0.0833$   | $0.8552 \pm 0.0111$   |
| KALS         | $0.7218 \pm 0.0087$   | $0.5607 \pm 0.0098$   | $0.8411 \pm 0.0095$   |
| KGMA         | $\mathbf{0.7361 \pm 0.0096}$ | $\mathbf{0.5762 \pm 0.1040}$ | $\mathbf{0.8673 \pm 0.0087}$ |

We verified the proposed algorithm using three kernels: projection kernel $k^{[proj]}$, canonical correlation kernel $k^{[CC]}$, and combined kernel $k^{[proj+CC]}$. In some cases, $k^{[proj]}$ is better than $k^{[CC]}$ or vice versa, suggesting that the kernels combination is more suitable

for different data distributions. For $k^{[proj+CC]}$, the mixing coefficients $\delta^{[proj]}$ and $\delta^{[CC]}$ were fixed at one. We obtain better results by combining $\delta^{[proj+CC]}$ two kernels.

*5.2. Datasets*

Three datasets were used in the experiments: JHMDB [44], HMDB51 [45], and UCF101 [46]. The **JHMDB** dataset has 21 action categories. The average recognition accuracies over three training–test splits are reported. The **HMDB51** dataset records 51 action categories. We reported the MAP over three training–test splits. The **UCF101** dataset includes 101 action categories, containing 13,320 video clips. The average accuracy of the first split was reported.

For the JHMDB dataset, we followed the standard data partitioning (three splits) provided by the authors. For other datasets, we used the first split provided by the authors, and applied the original testing sets for fair comparison. Because the semi-supervised training set contained unlabeled data, we performed the following procedure to reform the training set for each individual dataset. The class number $c$ was denoted for each dataset ($c$ = 21, 51, and 101 for JHMDB, HMDB51, and UCF101, respectively).

Using JHMDB as an example, we first randomly selected 30 training samples per category to form a training set ($30 \times c$ samples) in our experiment. From this training set, we randomly sampled $m$ videos ($m$ = 3, 5, 10, and 15) per category as labeled samples. Therefore, if $m = 10$, $10 \times c$-labeled samples will be available, leaving ($30 \times c - 10 \times c$) videos as unlabeled samples for the semi-supervised training setting. We used three splits of testing set on JHMDB and HMDB51 but only the first testing split on UCF101 due to lack of GPU memory resources. Owing to the random selected training samples, the experiments were repeated 10 times to avoid bias.

*5.3. Experimental Setup*

To demonstrate the superiority of our approach (KGMA), we adopted 8 methods for comparison: SVM, SFUS [47], SFCM [9], MFCU [10], KSPG, and KALS. Notably, SFUS, SFCM, MFCU, KSPG, and KALS are semi-supervised action recognition approaches. Using the available codes, we can facilitate a fair comparison.

For the semi-supervised parameters $\eta, \beta, \mu$ for SFUS, SFCM, MFCU, KSPG, KALS, and KGMA, we follow the same settings utilized in [9,10], ranging from $\{10^{-4}, 10^{-3}, 10^{-2}, 10^{-1}, 1, 10^1, 10^2, 10^3, 10^4\}$. Because the PBB parameters were not sensitive to our algorithm, we initialized the parameters as in [20], as indicated in Algorithm 1. Notably, since KGMA applied PBB to solve the optimal value of the objective function (8), it resulted in non-monotonic convergence with oscillating objective function values, as shown in Figure 4. Thus, using only the absolute error made it difficult to determine when to stop iterating, and relative error of objective function values was better than absolute error, which may be mathematically improper convergence. We chose constant $\varepsilon = 10^{-4}$ as the iteration-stopping criterion in (9).

*5.4. Mathematical Comparisons*

The recognition results with handcrafted features on three datasets are demonstrated in Figure 3. We compared our method with deep-learned features in Table 1.

Regarding the presented objective function (8), Figure 4 summarizes the computational results of the three optimization methods. When we used the 2048-dimensional deep-learned features by TSN on the JHMDB dataset, the model was trained with only 15 labeled samples and 15 unlabeled samples per class. With the same semi-supervised parameters set up, $\eta, \beta, \mu$, the performance differences during the solving of the same objective function could be compared in terms of running time, number of iterations, absolute error, relative error, and objective function value. Figure 4 shows the convergence curves of the three optimization methods. Since both SPG and PBB were non-monotonic optimization methods with relatively large fluctuations in objective function values, we omitted the first 29 iterations of SPG and PBB in Figure 4, and only displayed the data

starting from the 30th iteration so as to better illustrate the monotonic convergence process of ALS.

As shown in Figure 3, for a randomly selected video data sample, ALS exhibited the fewest iterations, shortest running time, and fastest computation speed of 0.1220 s after extracting the deep features by TSN. In contrast, PBB exhibited the most iterations, longest running time, and slowest computation speed of 0.4212 s, while SPG's performance was intermediate between ALS and PBB. Considering Figures 4 and Table 2, it is evident that despite using the PBB optimization method, our KGMA algorithm still achieves the highest accuracy on the kernelized Grassmann manifold space. Nevertheless, Equation (9) using SPG results in marginal improvement over ALS, which is likely attributable to our novel kernelized Grassmann manifold space.

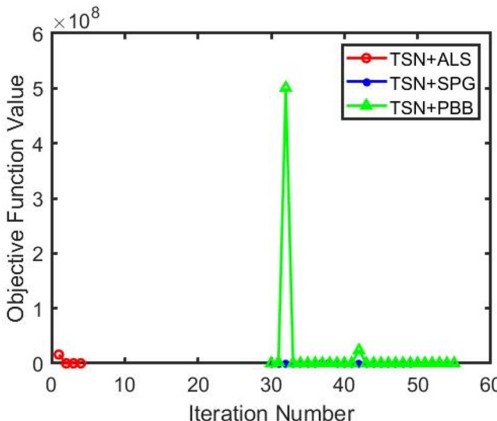

**Figure 4.** The convergence curves of the three optimization methods on the JHMDB dataset, with the final convergence results shown in Table 2. Due to the larger oscillations of PBB, the data for the first 29 iterations of SPG and PBB have been omitted here in order to better illustrate the comparative convergence of ALS, SPG and PBB.

**Table 2.** Mathematical results on JHMDB using $15 \times c$-labeled training samples. "Obj-Val" means objective function value.

| Methods | Features (dim $\times$ nSample) | Parameters | Times (s) | Iter. | Error | Relative Error | Obj-Val |
|---|---|---|---|---|---|---|---|
| ALS | TSN ($2048 \times 660$) | $\eta = 0.001, \beta = 0.01, \mu = 0.001$ | 0.4880 | 4 | 0.5972 | $2.0691 \times 10^{-4}$ | 2.0137 |
| SPG | TSN ($2048 \times 660$) | $\eta = 0.001, \beta = 0.01, \mu = 0.001$ | 6.1992 | 49 | 0.4706 | $8.1024 \times 10^{-4}$ | 32.0130 |
| PBB | TSN ($2048 \times 660$) | $\eta = 0.001, \beta = 0.01, \mu = 0.001$ | 23.5855 | 56 | 0.6146 | $7.1873 \times 10^{-4}$ | 10.0185 |

### 5.5. Performance on Action Recognition

A linear SVM was utilized as the baseline. Based on the comparisons, we observe the following: (1) KGMA achieved the best performance, and our semi-supervised algorithm was better than linear SVM, which is a widely used supervised classifier (2) all methods achieved better performances using more labeled training data, as shown in Figure 3, or enlarging the semi-supervised parameter (i.e., $\eta, \beta, \mu$) range such as Figure 5; (3) we averaged an accuracy of $3 \times c$, $5 \times c$, $10 \times c$, and $15 \times c$ cases, and the recognition of KGMA on JHMDB, HMDB51, and UCF101 improved by 2.97%, 2.59%, and 2.40%, respectively. When using TSN features, the recognition of our KGMA on the above-mentioned datasets improved by 2.21%, 3.77%, and 2.23%, respectively. Evidently, our semi-supervised method can improve recognition by leveraging unlabeled data compared to linear SVM with labeled data merely. Figure 3 illustrates that our algorithm benefits from the multiple-feature analysis, kernelized Grassman space, and iterative skills of the PBB method.

These results can be attributed to several factors. First, our method not only leverages semi-supervised approaches, but also leverages intra-class action variation and inter-

class action ambiguity simultaneously. Therefore, ours gain more significant performance than other approaches when there are few labeled samples. Second, we uncover the action feature subspace on the Grassmannian manifold by incorporating Grassmannian kernels, and solve the objective function optimization by the adaptive line-search strategy and the PBB method mathematically. Hence, the proposed algorithm works well in few labeled cases.

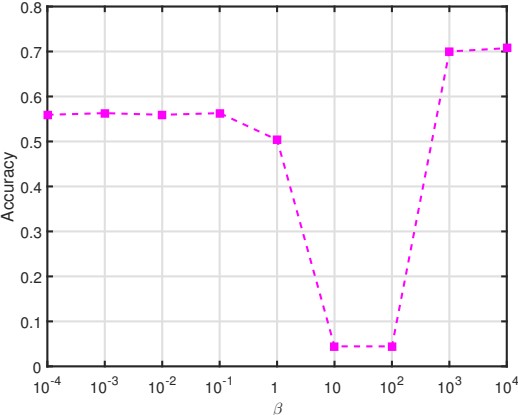

**Figure 5.** Accuracy on JHMDB using TSN, w.r.t the parameter $\beta$ with fixed $\eta$ and $\mu$.

*5.6. Convergence Study*

According to the objective function (4), we conducted experiments with the TSN feature, fixed the semi-supervised parameters $\eta, \beta, \mu$, and then executed both the ALS and PBB methods 10 times. The results of the study are listed in Table 2. Although no oscillation exists in the convergence of the ALS and it requires fewer iterations, the PBB method can outperform the ALS for three reasons. First, the PBB method uses a non-monotone line-search strategy to globalize the process [21], which can obtain the global optimal objective function value rather than being trapped in local optima using the monotone ALS method. Second, the character of adaptive step sizes is an essential characteristic that determines efficiency in the projected gradient methodology [21], whereas the iteration step skill has not been considered in ALS. Finally, the efficient convergence properties of the projected gradient method have been demonstrated because the PBB is well defined [21].

*5.7. Computation Complexity*

In the training stage, we computed the Laplacian matrix $L$, the complexity of which was $O(n^2)$. To optimize the objective function, we computed the projected gradient and trace operators of several matrices. Therefore, the complexity of these operations was $O(n^3)$.

*5.8. Parameter Sensitivity Study*

We verified that KGMA benefits from the intra-class and inter-class by manifold discriminant analysis, as shown in Figure 5. We analyze the impact of manifold learning on JHMDB and HMDB51, set $\eta = 10^3$ and $\mu = 10^{-1}$ at optimal values over split2, for $15 \times c$-labeled training data. As $\beta$ varied from $10^{-4}$ to $10^4$, the accuracy oscillated significantly and reached a peak value when $\beta = 10^4$. Since $\beta$ controls the proportion of the intra-class local geometric structure and the inter-class global manifold structure, as shown in Figure 5, when the intra-class local geometric structure is treated as a constant 1, $\frac{\beta}{1}$ can be considered such that the inter-class global manifold structure has a larger proportion in the objective function and vice versa. When $\beta = 0$, no inter-class structure is utilized; thus, if $\beta \to +\infty$, no intra-class structure is present. When the Grassmann manifold space leverages an adequate balance of intra-class action variation and inter-class action ambiguity, the proposed algorithm can further enhance the discriminatory power of the transformation matrix.

## 6. Conclusions

This study proposed a new approach to categorize human action videos. With Grassmannian kernel combinations and multiple-feature analysis on multiple manifolds, our method can improve recognition by uncovering the intrinsic features relationships. We evaluated the presented approach on three benchmark datasets, and experiment results show ours outperformed all competing methods, particularly when there are few labeled samples.

**Author Contributions:** Conceptualization, Z.X.; methodology, Z.X., X.L., J.L. and H.C.; software, Z.X., X.L. and J.L.; validation, Z.X.; formal analysis, Z.X. and X.L.; investigation, Z.X. and H.C.; resources, Z.X. and R.H.; data curation, Z.X.; writing—original draft, Z.X., X.L., J.L. and H.C.; writing—review and editing, X.L. and J.L.; visualization, Z.X.; supervision, R.H.; project administration, Z.X.; funding acquisition, Z.X. All authors have read and agreed to the published version of the manuscript.

**Funding:** This work was supported by the National Natural Science Foundation of China (61862015, 11961010, 12261026), the Science and Technology Project of Guangxi (AD21220114), the Guangxi Key Laboratory of Automatic Detecting Technology and Instruments (YQ23103), the Outstanding Youth Science and Technology Innovation Team Project of Colleges and Universities in Hubei Province (T201923), the Key Science and Technology Project of Jingmen (2021ZDYF024), the Guangxi Key Research and Development Program (AB17195025).

**Institutional Review Board Statement:** Not applicable.

**Informed Consent Statement:** Not applicable.

**Data Availability Statement:** Not applicable.

**Acknowledgments:** Many thanks to all the authors who took the time out of their busy schedules to review the paper and provide references.

**Conflicts of Interest:** The authors declare no conflict of interest.

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
