# Peer review of "Action Recognition via Adaptive Semi-Supervised Feature Analysis"

_applsci, doi:10.3390/app13137684_

Round 1

Reviewer 1 Report

In general, this is a good manuscript with interesting content. The overall structure of it is logical and good to follow. The related work section provides details, where it is needed. I particularly liked the methodical section and mathematical details, which are consistent and seem technically sound.

Nevertheless, before publication there are some issues that need to be worked on. Especially, the overall presentation and the content of the experimental section is not very convincing:

(1) Introduction needs in improvement

The introduction is already very technical with a lot of details, which makes it hard on the reader to understand the need of the proposed work. There is no consistent story and motivation. It is also rather short. I recommend to add more text to both reduce density of information and guide the reader towards the core aspects of the paper.

(2) Citation needs improvement

If found it very odd and rather alarming that datasets, which were used in this study, are not properly cited. Instead an IJCV article (Wang et al. 2016) is cited, that made also use of them. This is not sufficient! Please, cite the original papers that introduced those datasets.

In general, the amount of cited works is small for a journal article (34). Without checking references in detail, the previous point leaves me little worried whether all important work in this matter is cited.

(3) Experimental section needs improvement

There are only average precisions presented for the chosen datasets. A more detailed analysis (good cases, bad cases, etc.) would be necessary to better understand strengths and weaknesses of the proposed method.

Additionally, the authors present assumptions on why the performance increased (lines 282-288), but there is no evidence using experiments testing those individual assumptions.

It is not visible based on the results, that the suggested method works especially good with few labeled samples as is claimed in the conclusion (lines 323f) and introduction. The ordering and relative distances between methods stay the same (see Fig. 2, a-d).

Finally, please add subsections to this section to make it more readable!

(4) Overall visual presentation can be improved

While the overall structure is good, the visual details (especially figures) need some work. Figures 1 and 2 have very bad resolution. Please consider creating vector graphics. Additionally, Fig. 1 doesn't show the method very well. The important parts are way too small, while there is a lot of redundancy in other parts of that figure. For a better comparison in Fig. 2 all y-axes should have the same scaling (e.g until 0.8). Finally, the use of Fig. 3 is very limited at the moment. Authors should at least use a logarithmic scale for the y-axis to make differences in smaller function values better visible.

(5) Other issues
    - line 186 is out of bounds
    - Notation in reference section is inconsistent (author names sometimes in capital letters; first name and last name switched; writing style of journal names; etc.)

Most of the paper is convincing, but before these mentioned issues are not improved on, I can't recommend an immediate publication.

The writing / grammar needs to be improved. In general, there are some weird sounding sentences. Especially hard to read, broken or too long are the following sentences: lines 16-17, 53-57, 61-64, 100-105

Author Response

We have solved those issues according to the reviewer's suggestions.

Reviewer 2 Report

The manuscript presents a semi-supervised action recognition method using a adaptive feature analysis.  The work introduces the Grass-mannian kernels and Projected Barzilai-Borwein (PBB) method to train a subspace projection matrix as a classifier. Authors performed experiment that show good results as compared to existing approaches. The authors claimed following contributions: 1) simultaneous consideration of manifold learning and Grassmannian kernels in semi-supervised action recognition both for inter-class and intra-class separability, 2) the proposed PBB model for improved performance, 3) Experimentation to validate the proposed work. 

The document is well written and organized. However, small observations may be considered to improve the readability:

As mentioned above, the authors claimed 3 contributions. Normally, in all kind of research, a model is proposed and validated through experimentation results. So, the authors should explain how did they claim the contribution 1 and 3. 

The author should explain (may be with some example/scenarios), how the proposed model can be used for inter- and intra class separability?

The authors should given explanation of the datasets; their availability, features, quality, etc.

The applied side of the work should also be mentioned?

In what type of actions the  projection kernel, canonical correlation kernel, and combined kernel should be preferred. It means, how it would be decided to use which type of kernel?

Some punctuations are missing.

Author Response

(The authors gave the same response as above.)

Round 2

Reviewer 1 Report

The authors addressed most of my suggestions and concerns. I'm still not fully convinced about the claim, that the method is especially good when using few exampes. Also, in my opinion Figure 4 isn't really worth showing. However, these small personal issues do not warrant a further delay in publication. I would only ask the authors to carefully double-check section 1 again to fix some sentences and formulations (e.g. sentences in lines 18-21, 22-23).

English is mostly acceptable, but it is sometimes hard to understand the message (see other comments on introduction section).